# Regulation of Neuronal Chloride Homeostasis by Pro- and Mature Brain-Derived Neurotrophic Factor (BDNF) via KCC2 Cation–Chloride Cotransporters in Rat Cortical Neurons

**DOI:** 10.3390/ijms25116253

**Published:** 2024-06-06

**Authors:** Mira Hamze, Cathy Brier, Emmanuelle Buhler, Jinwei Zhang, Igor Medina, Christophe Porcher

**Affiliations:** 1INMED, INSERM, Aix-Marseille University, 13273 Marseille, France; mira.hamze@inserm.fr (M.H.); cahty.brier@inserm.fr (C.B.); emmanuelle.buhler@inserm.fr (E.B.); igor.medyna@inserm.fr (I.M.); 2INSERM (Institut National de la Santé et de la Recherche Médicale), Unité 1249, Parc Scientifique de Luminy, 13273 Marseille, France; 3INMED (Institut de Neurobiologie de la Méditerranée), Parc Scientifique de Luminy, 13273 Marseille, France; 4Shanghai Institute of Organic Chemistry, Chinese Academy of Sciences, 345 Ling Ling Road, Shanghai 200032, China; jinweizhang@sioc.ac.cn

**Keywords:** GABA, KCC2 activity, chloride homeostasis, BDNF

## Abstract

The strength of inhibitory neurotransmission depends on intracellular neuronal chloride concentration, primarily regulated by the activity of cation–chloride cotransporters NKCC1 (Sodium–Potassium–Chloride Cotransporter 1) and KCC2 (Potassium–Chloride Cotransporter 2). Brain-derived neurotrophic factor (BDNF) influences the functioning of these co-transporters. BDNF is synthesized from precursor proteins (proBDNF), which undergo proteolytic cleavage to yield mature BDNF (mBDNF). While previous studies have indicated the involvement of BDNF signaling in the activity of KCC2, its specific mechanisms are unclear. We investigated the interplay between both forms of BDNF and chloride homeostasis in rat hippocampal neurons and in utero electroporated cortices of rat pups, spanning the behavioral, cellular, and molecular levels. We found that both pro- and mBDNF play a comparable role in immature neurons by inhibiting the capacity of neurons to extrude chloride. Additionally, proBDNF increases the endocytosis of KCC2 while maintaining a depolarizing shift of E_GABA_ in maturing neurons. Behaviorally, proBDNF-electroporated rat pups in the somatosensory cortex exhibit sensory deficits, delayed huddling, and cliff avoidance. These findings emphasize the role of BDNF signaling in regulating chloride transport through the modulation of KCC2. In summary, this study provides valuable insights into the intricate interplay between BDNF, chloride homeostasis, and inhibitory synaptic transmission, shedding light on the underlying cellular mechanisms involved.

## 1. Introduction

KCC2 (Potassium–Chloride Cotransporter 2) and NKCC1 (Sodium–Potassium–Chloride Cotransporter 1) are crucial for maintaining chloride ion balance inside and outside neurons, playing complementary roles in the regulation of GABAergic (gamma-aminobutyric acid) inhibition and chloride homeostasis in the nervous system. NKCC1 intrudes chloride ions (Cl^−^), leading to a higher intracellular chloride concentration ([Cl^−^]_i_), while KCC2 actively extrudes Cl^−^, establishing a negative Cl^−^ gradient across the cell membrane [1,2]. The Cl^−^ gradient created by KCC2 is essential for GABAergic inhibition as it determines the direction of chloride flow through ionotropic GABA_A_ receptors (GABA_A_R). In mature neurons with functional KCC2, the low [Cl^−^]_i_ maintained by the transporter ensures that the influx of Cl^−^ ions through GABA_A_Rs hyperpolarizes the neuron, resulting in inhibition [1,3]. This mechanism dampens neuronal excitability. During early brain development, NKCC1 expression is predominant, resulting in higher [Cl^−^]_i_ in immature neurons. Consequently, GABAergic receptor activation leads to depolarizing responses and contributes to the excitatory actions of GABA during this critical period [1,4]. These depolarizing responses are essential for synaptic plasticity and the refinement of neural circuits in response to environmental stimuli. As the nervous system matures, there is a developmental shift in the relative functional expression of NKCC1 and KCC2. This shift is vital for the transition from depolarizing to hyperpolarizing responses to GABA. Increased KCC2 expression promotes Cl^−^ extrusion from neurons, reducing [Cl^−^]_i_ and strengthening inhibitory signaling [5,6,7]. This transition establishes an appropriate balance between excitation and inhibition, thereby contributing to the closure of the critical period and normal brain functioning [8]. Changes in the expression or functionality of NKCC1 and KCC2 are associated with a range of neurological disorders. These conditions include neurodevelopmental disorders (NDDs) like epilepsy, autism spectrum disorder (ASD), schizophrenia, Down syndrome, and Rett syndrome [9,10] and, more recently, neurodegenerative diseases [11,12]. All these conditions have been linked to disruptions in the expression or function of these cotransporters. Dysregulated NKCC1 or impaired KCC2 maturation can disrupt chloride homeostasis, leading to neuronal hyperexcitability and altered synaptic plasticity. The expression patterns of NKCC1 and KCC2 are tightly regulated by various factors, including neurohormones, hormones, trophic polypeptides, and neurotrophins such as BDNF (brain-derived neurotrophic factor) [13,14,15]. BDNF exists in two forms: proBDNF (the precursor of mature BDNF) and mature BDNF (mBDNF). These two forms of BDNF interact with different receptors and elicit distinct cellular responses [16]. In the neurons of rodents, the expression of proBDNF and p75 neurotrophin receptor (p75^NTR^) undergoes developmental regulation, with the highest levels observed in the early postnatal weeks, followed by a reduction at 6 weeks after birth [17]. This developmental pattern correlates with increased [Cl^−^]_i_ and the depolarizing action of GABA. In contrast, mBDNF is minimally detectable during the first postnatal week but reaches full expression around the third postnatal week [18,19]. These two forms of BDNF play distinct roles and have the potential to interact with KCC2. However, the intracellular pathways involving BDNF receptors and KCC2 remain unclear [15]. Several studies have suggested that proBDNF can reduce KCC2 protein expression and impair its function in neurons. This downregulation of KCC2 disrupts chloride ion homeostasis, leading to impaired inhibitory neurotransmission. Consequently, proBDNF promotes neuronal hyperexcitability and can contribute to conditions associated with reduced inhibitory signaling, such as epilepsy [20,21]. On the other hand, mature BDNF has been shown to either upregulate KCC2 expression and enhance its activity in immature neurons through the BDNF-TrkB-ERK1/2 (extracellular signal-regulated kinase 1/2) signaling pathway [14,22] or reduce KCC2 function in mature neurons via the BDNF-TrkB-PlC pathway [13]. This dual effect of mature BDNF can promote seizure susceptibility and contribute to neuropathic pain [13,23,24]. These findings collectively support the notion that mBDNF, proBDNF, and KCC2 interact at different stages of brain development to regulate neuronal survival, synaptic development and plasticity, and the balance between excitation and inhibition. The dysregulation of this interplay, such as an imbalance between proBDNF and mature BDNF or alterations in KCC2 expression or function, has been implicated in various neurological disorders and conditions. Therefore, understanding the intricate mechanisms underlying their interactions is crucial for unraveling the role of these factors in normal brain function and their implications for neurological disorders. In this study, we aimed to investigate the respective roles of both forms of BDNF in regulating chloride homeostasis and GABAergic inhibitory strength during early developmental stages. We achieved this by examining the functional expression of KCC2 in cultured rat hippocampal neurons. Additionally, we explored the behavioral consequences of proBDNF expression using electroporated rat pups.

## 2. Results

### 2.1. Functional Activity of KCC2 and BDNF Signaling Pathways in Maturing Cultured Hippocampal Neurons

The developmental excitatory–inhibitory GABA sequence mediated by the upregulation of KCC2 is closely paralleled by the downregulation of proBDNF and p75^NTR^ expression in cortical regions [4,18]. Furthermore, an interplay between depolarizing GABA, the KCC2/NKCC1 imbalance, and p75^NTR^ has previously been shown after neuronal injury [25]. These observations led us to explore whether the developmental expression of KCC2 was present in our primary cultured hippocampal neurons. We initially investigated the developmental expression of KCC2 by measuring KCC2 fluorescence intensity from 1 day in vitro (DIV) to 9 DIV. Our findings revealed a very low expression of KCC2 from 1 DIV to 3 DIV, succeeded by a swift and sustained escalation in KCC2 fluorescence intensity from 5 DIV to 9 DIV (Figure 1A,B). Next, our objective was to confirm the functionality of the TrkB signaling pathway at 6–7 DIV, induced by the exogenous application of mature BDNF (mBDNF), in comparison to cleavage-resistant proBDNF (CR-proBDNF) application and control conditions. We quantified the ratio of the phosphorylated form of the TrkB receptor (pTrkB) to MAP2 intensity in primary cultured hippocampal neurons (Figure 1C,D). This was achieved by utilizing primary antibodies targeting the phosphorylated form relative to the total expression of MAP2 fluorescence intensity in cultures treated with CR-proBDNF (25 ng/mL, 2 h), mBDNF (25 nM, 2 h), TAT-Pep5 (2 µM, 2 h), or TrkB-IgG (1 µg/mL, 2 h). We observed a significant increase in the pTrkB/MAP2 ratio intensity in neurons treated with mBDNF compared to other conditions. (Mean values were 1.87 ± 0.99 a.u. for mBDNF vs. 1.01 ± 0.66 a.u. for CR-proBDNF, 0.82 ± 0.63 a.u. for TAT-Pep5, and 0.81 ± 0.44 a.u. for TrkB-IgG when compared to the control condition; see Figure 1C,D. Here, and throughout the results, all numbers indicate mean ± S.D. values.) These results further suggest the specificity of the signaling pathway activated by mBDNF compared to CR-proBDNF, as proBDNF failed to activate the TrkB pathway.

To assess the developmental activity of KCC2 in cultured hippocampal neurons, we performed the NH4^+^ flux assay. This assay offers an estimation of the ion transport efficacy of KCC2, providing valuable insights into its functionality across different developmental stages [26,27]. Changes in NH4^+^-dependent intracellular pH (pH_i_) were monitored using the ratiometric fluorescent probe [28] composed of pH-sensitive pHluorine [29] and pH-insensitive mCherry (Figure 2A,B). The advantage of this genetically encoded pH-sensitive probe (pH-sensor) is that it allows for visualization exclusively in transfected cells. When the NH4^+^-containing media were applied to hippocampal cultured neurons, there was a progressive acidification of the pH_i_ following a slight and brief (10–30 s) alkalinization event (Figure 2C,D). We observed higher rates of NH4^+^-dependent acidification in older (more mature) neurons (12–13 DIV) compared to their younger counterparts (6–7 DIV), indicating heightened KCC2 transport activity operating in reverse mode, facilitating the influx of NH4^+^ (mean values were −173 ± 107 a.u. at 6–7 DIV vs. −649 ± 207 a.u. at 12–13 DIV for the control condition; see Figure 2E). This effect was diminished in the presence of 70 µM of bumetanide, a diuretic that inhibits KCC2 at this concentration (mean values were −176 ± 114 a.u. at 6–7 DIV vs. −328 ± 164 a.u. at 12–13 DIV for the bumetanide condition; see Figure 2E). Overall, these results align with the progressive increase in KCC2 intensity and confirm the heightened activity of KCC2 during the early development of neuronal cell cultures.

### 2.2. proBDNF Maintains a Depolarized GABA Response in Hippocampal Neurons

To investigate the effects of BDNF on GABA polarity shift and neuronal chloride homeostasis, we performed the gramicidin-perforated patch-clamp technique to measure the GABA_A_ reversal potential (E_GABA_) and calculated the intracellular chloride concentration ([Cl^−^]_i_) in dissociated hippocampal cultures in immature neurons at 6–7 DIV and more mature neurons at 12–13 DIV. We measured E_GABA_ in neurons transfected with GFP (control), GFP-BDNF-Cherry (BDNF), or GFP-CR-proBDNF-Cherry (CR-proBDNF). Because BDNF has been shown to influence the activity of NKCC1 [30], expressed in neurons and astrocytes [31], all measurements of E_GABA_ were performed in the presence of bumetanide, an inhibitor of NKCC1 at low concentrations (10 μM). As previously described, in our preparations of cultured hippocampal neurons, 10 μM bumetanide produced a 5 mV negative shift of E_GABA_ in immature neurons (6–7 DIV) and an 8 mV negative shift in more mature cells (13–15 DIV) [32]. Our results revealed significant differences in E_GABA_ and intracellular chloride concentration ([Cl^−^]_i_) in both immature and mature neurons between the control condition and CR-proBDNF-transfected cells (mean values were −80.3 ± 8.54 mV and 6.9 ± 2.11 mM for the control vs. −72.1 ± 6.43 mV and 9.3 ± 2.4 mM for CR-proBDNF at 6–7 DIV and −85.3 ± 8.64 mV and 5.8 ± 1.85 mM for the control vs. −79.1 ± 3.5 mV and 6.9 ± 0.95 mM for CR-proBDNF at 12–13 DIV; see Figure 3C–F) and between CR-proBDNF- and BDNF-transfected cells (−81.3 ± 5.72 mV and 6.5 ± 1.42 mM at 6–7 DIV vs. −86.3 ± 7.2 mV and 5.4 ± 1.4 mM at 12–13 DIV for the BDNF condition; see Figure 3C–F). Conversely, E_GABA_ and [Cl^−^]_i_ in BDNF-transfected neurons showed no significant differences in both immature and mature neurons when compared to control conditions. The absence of significant differences between the control and BDNF conditions could be attributed to various factors. Firstly, BDNF-transfected neurons are likely to express both forms of BDNF (pro and mature), potentially obscuring the specific effects of mature BDNF alone. Additionally, it is important to consider the presence of trophic factors in the culture media, which could also influence the observed outcomes. Furthermore, these findings suggest that pro-BDNF may play a role in maintaining E_GABA_ and intracellular chloride concentration in an immature state.

### 2.3. Effects of Pro- and Mature-BDNF on KCC2 Cell Trafficking

Given that low [Cl^−^]_i_ is KCC2-dependent, we conducted a live-staining analysis to assess the surface expression and internalization of KCC2-pH_ext_. The stability of KCC2 at the cell surface is dependent on the process of phosphorylation or dephosphorylation of amino acid residues in the C-terminal part of the protein. For instance, the activation of Serine^940^ (Ser^940^) residue increases the stability of KCC2 in the plasma membrane, whereas the dephosphorylation of Ser^940^ or the phosphorylation of Threonine 906/1007 (Thr^906/1007^) residues alter surface expression abilities and promote KCC2 endocytosis [33,34]. We, therefore, assessed whether CR-proBDNF and BDNF could regulate KCC2 stability at the cell surface of hippocampal neurons by measuring KCC2 expression in different cell compartments at 9 DIV. As a tool, we used a KCC2 construct tagged in an external loop with a fluorescent protein pHluorine (KCC2-pH_ext_) [34]. As detailed in the Methods section, this construct allowed for the visualization and quantification of KCC2-pH_ext_ molecules that were decorated with a specific antibody on the surface of living neurons over two hours (F_all_), the number of molecules labeled and internalized during this period (F_i_), the amount of surface-expressed molecules at a given instant (F_m_), and the total amount of KCC2-pH_ext_ overexpressed by a neuron (F_t_) (Figure 4A–D). Before proceeding with the live staining of KCC2-pH_ext_, we co-transfected neuronal cultures at 7 DIV with KCC2-pH_ext_ and BDNF isoform constructs (CR-proBDNF and BDNF). As positive controls, we employed KCC2 mutant constructs to either retain KCC2 at the neuronal membrane (A/A KCC2 mutant) or prevent its targeting to the neuronal membrane (ΔNTD-KCC2 mutant) [35]. Our findings indicated a significant increase in all the clusters of KCC2 expression (F_all_) in neurons transfected with A/A or BDNF isoform constructs compared to the control group (mean values were 1 ± 0.5 a.u. for the control vs. 2.01 ± 1.24 a.u. for A/A, 2.14 ± 1.11 a.u. for CR-proBDNF, and 1.32 ± 1.04 a.u. for BDNF; see Figure 4A,B). As expected, the A/A KCC2 mutant condition led to an increase in the KCC2 membrane pool (mean values were 2.07 ± 1.61 a.u. for A/A vs. 1 ± 0.42 a.u. for the control; see Figure 4A,C). Regarding the ΔNTD-KCC2 mutant, the multi-step immunolabeling protocol did not detect any membrane expression (0.013 ± 0.014 a.u. for ΔNTD; Figure 4A,C) or internalization (0.01 ± 0.58 a.u. for ΔNTD vs. 0.99 ± 0.7 a.u. for the control; Figure 4A,D), which confirmed the integrity of the cell membrane during the experiment. In comparison to the control group, neurons transfected with the CR-proBDNF construct exhibited a significant increase in the amount of endocytosed KCC2-pH_ext_ (F_i_) (0.99 ± 0.7 a.u. for control vs. 2.7 ± 1.93 a.u. for CR-proBDNF; Figure 4A,D), while the amount of KCC2 internalization (F_i_) remained similar between the BDNF and control groups (1.73 ± 0.93 a.u. for BDNF; Figure 4A,D). In summary, our results revealed an elevated rate of KCC2 endocytosis in the presence of CR-proBDNF, underscoring the potential of this immature form of BDNF to trigger the increased internalization of KCC2.

### 2.4. KCC2-Dependent Ammonium Transport in BDNF-Treated Hippocampal Neurons

To further investigate whether treatment with both forms of BDNF affected the Cl^−^ extrusion capability of the KCC2 transporter, we conducted the NH4^+^ flux assay. As described before, this assay allows us to assess changes in ion transport efficacy, providing insights into the modulation of KCC2 function by BDNF. Given the potential masking effects of trophic factors present in the culture media or endogenously secreted BDNF by the cell cultures, in this set of experiments, we chose to block the p75^NTR^ and TrkB receptor signaling pathways to uncover the specific action of pro- and mBDNF. Hippocampal neuronal cultures were treated at 6–7 DIV with either the p75^NTR^ signaling inhibitor TAT-Pep5 or the mBDNF scavenger TrkB-IgG. These treatments were administered in the presence or absence of 70 µM of bumetanide to block the co-transporters NKCC1 and KCC2. Neurons treated with TAT-Pep5 showed a significantly faster NH4^+^-induced acidification rate compared to the control group (−484 ± 216.2 a.u. for TAT-Pep5 vs. −173 ± 107 a.u. for the control condition; Figure 5A,C). Notably, the presence of 70 µM of bumetanide effectively abolished this effect, underscoring the specificity of the p75^NTR^ signaling pathway’s impact on KCC2 (−201.5 ± 161.5 a.u. for + bumetanide 70 µM vs. −484 ± 216.2 a.u. for the TAT-Pep5 condition; Figure 5C). Interestingly, neurons treated with TrKB-IgG displayed a similar significant increase in acidification rate compared to the control (−291 ± 117 a.u. for TrkB-IgG vs. −173 ± 107 for the control; Figure 5B,D). This effect was blocked in the presence of 70 µM of bumetanide (−191 ± 77 a.u. for TrkB-IgG + bumetanide 70 µM vs. −291 ± 117 a.u. for TrkB-IgG; Figure 5D). Taken together, these findings suggest that during this specific developmental stage (6–7 DIV), pro- and mBDNF serve a comparable function through their respective receptors, p75^NTR^ and TrkB, in diminishing the neuronal cell’s capacity to extrude chloride, thereby resulting in the maintenance of a depolarizing E_GABA_.

### 2.5. CR-proBDNF-Electroporated Rat Pups Exhibited Behavioral Deficits 

Building upon the data obtained from our in vitro experiments, our investigation was expanded to in vivo analysis, specifically examining the developmental milestones of in utero electroporated (IUE) rats with either GFP or CR-proBDNF or BDNF constructs. Given the known role of BDNF in regulating energy balance and promoting anorectic signals [36], our initial focus was to analyze the weight of electroporated rats during the first two postnatal weeks. No weight loss was observed in rats electroporated with either CR-proBDNF or BDNF (Figure 6A). Furthermore, IUE with BDNF constructs did not show a significant impact on other physical landmarks, such as eye opening, incisor eruption, and fur development (Figure 6B). In addition, we aimed to explore meaningful behaviors indicative of cortical circuit maturation. We observed huddling behavior (as depicted in Figure 6C,D), a valuable model for studying early social interaction with peers [37,38]. We fitted our results in a non-linear regression fit model (R^2^ = 0.9184 for GFP, R^2^ = 0.9047 for CR-proBDNF, R^2^ = 0.9066 for BDNF; Figure 6D). Despite no observed difference between the control group (GFP) and BDNF-electroporated rodents, we noted a significant decrease in huddling behavior in rats electroporated with CR-proBDNF when compared to both the control group and the BDNF groups (Figure 6C,D). In summary, these findings demonstrate that the overexpression of CR-proBDNF in the somatosensory cortex leads to sensory deficits, as evidenced by the results of the huddling test.

### 2.6. CR-proBDNF-Electroporated Rats Exhibited Deficits in Sensory–Motor Maturation

To evaluate the influence of CR-proBDNF or BDNF overexpression in the somatosensory cortices of IUE rats on the emergence of developmental reflexes, we conducted the righting reflex test. Overall, the results revealed that rat pups electroporated with BDNF plasmid displayed a reduced time to correct positioning compared to the control group between postnatal ages 8 (PN 8) and 10 (PN 10) (mean values 1.99 ± 0.55, 1.97 ± 0.76, and 1.56 ± 0.41 s for the control vs. 1.57 ± 0.6, 1.37 ± 0.48, and 1.34 ± 0.53 s for BDNF between PN 8–10; Figure 7A) and to CR-proBDNF rodents at PN 9 (mean values 1.37 ± 0.48 s for BDNF vs. 1.73 ± 0.57 s for CR-proBDNF; Figure 7A). Additionally, we found that CR-proBDNF exhibited a reduced time at PN 13 when compared to the control group (mean values 1.02 ± 0.076 s for CR-proBDNF vs. 1.14 ± 0.21 s for the control; Figure 7A). To further investigate potential sensory deficits in electroporated rats, we employed the cliff avoidance test. The results revealed that rat pups electroporated with the CR-proBDNF construct exhibited lower success rates in completing the test and spent more time turning away or retracting from the edge compared to both the GFP control group and rodents electroporated with BDNF (refer to Table 1 and Table 2 for cliff avoidance time and score; Figure 7B,C). These findings suggest an alteration in sensory–motor maturation in the CR-proBDNF condition.

## 3. Discussion

In altricial rodents, neonates’ brains display a depolarizing GABA activity that is particularly evident during the initial postnatal days in vivo or the first two postnatal weeks in vitro. This efflux of chloride (Cl^−^) ions is crucial for spontaneous network activity, such as giant depolarizing potentials, which play a role in the maturation of functional networks [4]. Consequently, during this early developmental period, chloride homeostasis shifts from an initially elevated intracellular chloride concentration ([Cl^−^]_i_) to lower [Cl^−^]_i_. This shift subsequently changes GABA polarity from depolarizing to inhibitory. Chloride homeostasis is primarily regulated by electroneutral secondary active co-transporters, NKCC1 and KCC2, functioning in opposing directions. Additionally, it is influenced by both sodium (Na^+^)-dependent and Na^+^-independent exchangers [39]. Thus, in healthy mature neurons, KCC2 activity dynamically regulates the constant cytoplasmic low levels of chloride ions. In this study, our primary objective was to elucidate how the principal brain neurotrophic factor (BDNF) regulates the expression and function of KCC2 during the early developmental period. Our primary findings, as observed in hippocampal neuronal cultures, demonstrate that proBDNF maintains a positive E_GABA_ in both mature and immature neurons. Additionally, the analysis of KCC2 trafficking correlated with electrophysiological data, revealing an increased endocytosis rate in CR-proBDNF-expressing neurons. Interestingly, the developmental period of GABAergic maturation during the first few postnatal weeks, transitioning from depolarizing to inhibitory responses, coincides with a surge in proBDNF [17,18] and low KCC2 expression [20]. In contrast, BDNF, acting through the TrkB signaling pathway, promotes the functional maturation of KCC2 [14,22]. Consequently, it is conceivable that both forms of BDNF play opposing roles in KCC2 expression and function in maturing neurons. The function of KCC2 is primarily dependent on the trafficking and targeting processes, which are intricately governed by post-translational regulations involving the (de)phosphorylation of specific amino acid residues. As mentioned earlier, Ser^940^ [33] and Thr^1007^ [32,40] have undergone extensive study, emphasizing their crucial roles in the regulation of surface stability and internalization, respectively. Our in vitro results did not demonstrate the capacity of both forms of BDNF to increase the cell surface expression of KCC2. However, proBDNF enhanced KCC2 turnover, implying a mechanism that could potentially constrain the functional efficacy of KCC2 while favoring NKCC1 activity. This finding highlights a regulatory dynamic wherein proBDNF may exert a modulatory influence on chloride homeostasis, potentially impacting neuronal excitability and synaptic transmission. Further investigations into the specific molecular mechanisms underlying this effect will be crucial for a comprehensive understanding of how proBDNF contributes to the intricate balance of chloride ion regulation in the brain. Moreover, a recent study revealed distinct membrane dynamics for NKCC1 compared to KCC2. NKCC1 tends to be more localized in endocytic zones rather than the membrane and exhibits a rapid transition from endocytic zones to the membrane [41]. The interplay of these various effects may contribute to a reversal of chloride flux in pathological conditions. Both forms of BDNF activate distinct signaling pathways. ProBDNF activates the RhoA-Rock pathway through p75^NTR^ [5,20,42], while mature BDNF activates the ERK1/2 or PLC signaling cascade through TrkB [22,43]. To gain a comprehensive understanding of the roles played by pro- and mBDNF in regulating downstream targets affecting the cell surface expression or stability of KCC2, further experimentation is warranted. It is established that the cell surface expression of KCC2 relies on the phosphorylation status of the KCC2 Thr^1007^ residue, a target of phosphorylation by WNK1. Several studies, particularly in hippocampal neuronal cultures in vitro, have demonstrated that the phosphorylation of these amino acids by the WNK1 pathway leads to the inhibition of KCC2 transport activity and increased internalization of the protein [32,44,45]. However, the precise contribution of mBDNF in facilitating the maturation of Cl^−^ homeostasis and GABA polarity through the potential downregulation of WNK1 remains to be elucidated. WNK1 is a pivotal player involved in KCC2 endocytosis and NKCC1 activation [46]. Complementary investigations are essential to unravel the intricate interplay between mBDNF signaling and the regulation of WNK1 activity, providing insights into the mechanisms governing chloride homeostasis and GABAergic signaling maturation. Additionally, the complexity of this regulatory mechanism is compounded by the potential involvement of proBDNF/p75^NTR^ in facilitating the mBDNF/TrkB signaling pathway [47]. This also suggests a scenario where p75^NTR^ and TrkB may collaboratively work to regulate the transcriptional and post-translational amplification of the mBDNF/TrkB pathway. This added layer of intricacy significantly contributes to regulating the expression and function of KCC co-transporters. At the cognitive level, both proBDNF and mBDNF are implicated in various neurological disorders and play a role in modulating learning and memory processes. For example, mBDNF has been shown to enhance synaptic transmission and promote synaptic plasticity, which is essential for optimal learning and memory [48,49,50]. Conversely, increased levels of proBDNF could potentially impede synaptic plasticity, thereby hindering memory consolidation and impacting cognitive functions, ultimately leading to deficits in behavioral tasks [51]. Importantly, these effects are age-dependent. The disruption of proBDNF signaling during the postnatal period has been shown to lead to disrupted spatial memory consolidation and a reduction in spine density [17]. The role of KCC2 and NKCC1 in maintaining chloride homeostasis extends beyond brain development, as dysfunctions of these co-transporters are also observed in various neurodegenerative disorders, including Alzheimer’s disease (AD), amyotrophic lateral sclerosis, Parkinson’s disease, and Huntington’s disease [10]. Notably, in Alzheimer’s disease, alterations in KCC2 activity occur during the presymptomatic phase and are linked to cognitive deficits [11]. Overall, while the specific mechanisms linking KCC2 to neurodegenerative diseases are still being elucidated, the dysregulation of KCC2-mediated chloride homeostasis appears to be a common feature in various neurodegenerative conditions. In this study, our focus was on the developmental period, and, to understand the impact of both forms of BDNF in a more physiological context, we utilized in utero electroporation (IUE) as a method to introduce either proBDNF or BDNF into the somatosensory cortex of rat pups. This approach enabled us to examine the potential behavioral effects resulting from the targeted introduction of this neurotrophic factor. During the neonatal period, typically encompassing the first few weeks after birth, the somatosensory cortex undergoes rapid and intricate development, establishing the neural foundation for sensory perception and motor control. Studies involving mice lacking BDNF have revealed significant deficiencies in coordination and balance (ataxia) and sensory deficits, underlining the critical role of BDNF in these processes [52,53]. Additionally, the expression of BDNF in the cortex has been identified as crucial for motor learning [54] and plasticity [55]. In our study, the observed results aligned with cellular-level data, clearly illustrating the contrasting effects of both forms of BDNF. Specifically, within the CR-proBDNF condition, we noted a deficit in sensory–motor responses in huddling and cliff avoidance tasks. These results strongly suggest that an imbalance between pro- and mature BDNF can potentially alter the functional maturation of cortical circuits during this critical developmental period. 

## 4. Materials and Methods

### 4.1. Reagents and Treatments

Isoguvacine was purchased from Tocris Cookson (Bristol, UK). Bumetanide was purchased from Sigma (St-Louis, MO, USA). Human CR-ProBDNF and human mBDNF were purchased from alomone labs (Jerusalem, Israel). TAT-Pep5 was purchased from Millipore (Molsheim, France). TrkB-IgG was purchased from the R & D system (Minneapolis, MN, USA).

### 4.2. Primary Cultures of Rat Hippocampal Neurons 

Neurons from 18-day-old rat embryos were dissociated with trypsin and plated at 300,000 cells/mL in a minimal essential medium (MEM) with supplements, as previously described [56]. On days 7, 10, and 13 of culture (DIV), half of the medium was replaced with MEM containing 2% B27 (Invitrogen, Carlsbad, CA, USA) and 2 mM L-Glutamine. For electrophysiology and surface labeling, neurons were plated on 12 mm coverslips in 35 mm dishes coated with polyethyleneimine (10 mg/mL).

### 4.3. Transfections

Neurons were transfected using 300 µL Opti-MEM, 7 µL Lipofectamine 2000, 1.5 µL CombiMag (OZ Biosciences, Marseille, France) per µg DNA, and 1.5 µg DNAs of interest, following the previously described method [35]. After a 15 min incubation at room temperature (RT), the mix was added to the culture. Dishes were then placed on a magnetic plate for 2 h at 37 °C with 5% CO_2_. Transfection was ended by replacing 50% of the solution with a fresh medium. Cells were used 24–48 h post-transfection for immature neurons and 96 h post-transfection for mature neurons. Generally, one plasmid was used per well, except for co-transfection with KCC2-pH_ext_ (0.6 µg) and BDNF constructs (0.9 µg).

### 4.4. NH^+^_4_ Flux Assay

Primary hippocampal neurons were transiently transfected using Lipofectamine 2000 and CombiMag with the ratiometric fluorescent probe composed of pH-independent mCherry and pH-sensitive pHluorine (called a pH sensor) [28,29]. Coverslips were treated for 2 h with either TAT-Pep5 (2 µM) or TrKB-IgG (1 µg/mL) in the presence or absence of bumetanide 70 µM and for 1 h with bumetanide 70 µM alone. For recordings, coverslips were submerged in a recording chamber and perfused with an extracellular Hepes-buffered solution (HBS) containing (in mM) 150 NaCl, 2.5 KCl, 5 HEPES, 2.0 CaCl_2_, and 2.0 MgCl_2_, at pH 7.4 and supplemented with 0.45% glucose and bumetanide 10 µM. The ratiometric fluorescence of pHluorine/mCherry was measured using an epifluorescence imaging setup on an inverted Olympus IX71 microscope (Tokyo, Japan) with a FITC/CY3 Dualband ET Filterset and additional single-band filters. pH-sensitive pHluorine fluorescence (F480) was obtained with a 480/20 excitation filter and 520/40 emission filter, while pH-insensitive mCherry-KCC2 fluorescence (F577) was obtained with a 577/25 excitation filter and 645/75 emission filter. Fluorescence was sampled at 0.1 Hz using a CoolSNAPHQ CCD camera and MetaMorph software (Version 7.7.5.0, Molecular Devices, San Jose, CA, USA). Excitation lasted 100 ms for F480 and 50 ms for F577. Recordings were performed with a 40× objective (NA 0.6). Baseline fluorescence was acquired for 5 min, followed by a 6 min perfusion with 10 mM NH_4_Cl solution and 2–5 min washout imaging. The ΔF/F ratio was calculated from the images, and the acidification rate (ΔR/min) was determined by the change in ΔF/F values between 0.5 and 5 min during NH4^+^ presence. For further details, please refer to the article [28].

### 4.5. Gramicidin-Perforated Patch-Clamp Recordings

Patch-clamp recordings using gramicidin were conducted on primary hippocampal neurons transfected with constructs encoding GFP and BDNF-piTracer (GFP-BDNF-mCherry) or CR-proBDNF-piTracer (GFP-CR-ProBDNF-mCherry), 1 to 4 days post-transfection (6–7 or 12–13 DIV). Neurons were perfused with an external Hepes-buffered solution (HBS) containing 150 mM NaCl, 2.5 mM KCl, 5 mM HEPES, 2.0 mM CaCl_2_, and 2.0 mM MgCl_2_, with 0.45% glucose and osmolarity of 300 mOsm. For recording, the external HBS contained 10 µM bumetanide. Recording micropipettes (5 MΩ) were filled with a solution containing 140 mM KCl, 5 mM HEPES, and 20 µg/mL gramicidin. Isoguvacine (30 µM) was applied to recorded cells via a micropipette connected to a Picospritzer. Recordings were performed in voltage-clamp mode using an Axopatch-200A amplifier (Molecular Devices, San Jose, CA, USA) and pCLAMP acquisition software (Version 8, Molecular Devices, San Jose, CA, USA). Data were low-pass filtered at 2 kHz and acquired at 10 kHz. Isoguvacine responses were recorded at different voltages depending on neuron GABA reversal potential. Linear regression was used to calculate the voltage dependence of the isoguvacine responses, and the Nernst equation was used to calculate [Cl^−^]_i_.

### 4.6. Immunocytochemistry and Confocal Microscopy 

Hippocampal cultures were treated as described above and fixed in 4% PFA–sucrose, followed by PBS or HBS washes and incubation in 0.3 M glycine. Blocking was performed with 1% BSA, 5% goat serum, and 0.3% Triton X-100 to reduce nonspecific binding. Next, cultures were incubated overnight at 4 °C with rabbit anti-KCC2 (1:3000, US-K0120–07, US Biological, Salem, MA, USA) or rabbit anti-phosphoTrkB (1:500, ABN1381, Millipore, Molsheim, France) along with chicken anti-MAP2 (1:500, ab5392, Abcam, Cambridge, UK). Primary antibodies were visualized with goat anti-rabbit Alexa-488 (1:500, 4412, Cell Signaling, Danvers, MA, USA)) and donkey anti-chicken Alexa-647 (1:500, AP194SA6, Millipore, Molsheim, France). After washing, coverslips were mounted with Vectashield. Immunoreactivity was sequentially acquired using a laser-scanning confocal microscope (Olympus Fluorview-500, Tokyo, Japan) with ×63 oil-immersion objective and adjustments to avoid saturation. Images were processed with MetaMorph software (version 7.7.5.0, Molecular Devices, San Jose, CA, USA).

### 4.7. Surface Immunolabeling on Living Neurons and Analysis of KCC2-pH_ext_ Proteins

For the immunolabeling of KCC2-pH_ext_ proteins on living neurons, rabbit anti-GFP (1:250, Invitrogen, A-6455) antibodies were applied to neurons in culture media for 2 h at 37 °C, with 5% CO_2_. Neurons were then rinsed three times with HBS at room temperature (RT) and labeled with goat anti-rabbit Alexa-647 conjugated antibody (1:250, Invitrogen, A-21244) for 30 min at 13 °C at the surface where KCC2-pH_ext_ was located at the moment of the cooling down of the cells (F_m_, membrane fluorescence pool). After fixing in 4% PFA–sucrose at 4 °C, cells were permeabilized and blocked with a mixture of 0.3 M glycine, 0.3% Triton X-100, and 5% GS for 1 h at RT. To reveal all labeled clusters (F_all_, surface + internalized fluorescence pool), cells were incubated with goat anti-rabbit Alexa-555 conjugated antibody (1:400, Invitrogen, A-21428) for 1 h at RT. For visualization of the total pool of overexpressed KCC2-pH_ext_ (F_t_, total expression pool), cells were labeled with chicken anti-GFP antibody (1:500, 1020, Avès lab Davis, CA, USA) and goat anti-chicken Alexa-488 conjugated antibody (1:1000, A-11039, Invitrogen, Carlsbad, CA, USA) for 1 h each at RT. Internalized KCC2-pH_ext_ (F_i_, internalized fluorescence pool) was quantified post-hoc using MetaMorph software, considering only Alexa 555-positive molecules. Images were acquired with an Olympus Fluorview-500 confocal microscope (Tokyo, Japan) using a 60× oil immersion objective (NA1.4, zoom 2.5). Each Z-stack included 9 planes taken with a 0.7 µm distance between planes. Absolute fluorescence values were used for statistical analysis, normalized by their respective control-KCC2-pH_ext_ mean values.

### 4.8. In Utero Electroporation 

In utero injections and electroporations (IUE) were performed on embryos from timed pregnant rats at embryonic day 15, as previously described [20]. Briefly, pregnant rats received buprenorphine (0.03 mg/kg, Buprecare, Axience, Pantin, France) and Rimadyl (5 mg/Kg, Zoetis, Malakoff, France) followed by anesthesia with Isoflurane (4% for induction, then 2.5%) after 30 min. Embryos’ lateral ventricles were injected with Fast Green (2 mg/mL; Sigma, St. Louis, MO, USA) combined with DNA constructs encoding GFP or BDNF-piTracer or CR-proBDNF-piTracer [20]. Electroporation was carried out using 40 V voltage pulses (BTX ECM 830 electroporator Harvard Apparatus, Holliston, MA, USA) delivered via tweezer-type electrodes (Nepa Gene Co., Chiba, Japan) across the uterine wall. IUE was performed at E15, an active period for the radial and tangential migration of newborn neurons in the neocortex [57]. Successfully electroporated pups were selected postnatally based on GFP reporter fluorescence. Morphological analysis of electroporated tissues was conducted using fluorescent stereomicroscopy. No alterations in cortical layer position or morphology were observed in GFP, BDNF, and CR-proBDNF conditions. However, approximately 30% of electroporations failed due to the absence of transfected cells, resulting in exclusion from the study.

### 4.9. Developmental Landmarks and Behavioral Tests

For all behavioral tests, animals were first acclimated to the behavioral room for 30 min. Control (GFP), CR-pro-, and BNDF rats were assessed by physical landmarks of rat development, the cliff avoidance test, the huddling behavior test, and the righting reflex test.

Physical landmarks of rat development. The developmental milestones measured in this study included weight, eye opening, incisor eruption, and fur development [58].

Righting reflex evaluation (RR). We assessed the RR by initially placing unrestrained rodents on their backs (supine position). Then, we examined whether animals restored the RR (flipped to the prone position). We measured the time it took for the animals to right themselves. This process was repeated 3 times consecutively and we recorded the average time for each attempt. We also scored the behavior: 0 if the animal stayed on its back, 1 if laid on its side, and 2 if the animal flipped onto its belly. A maximum time of 30 s was given for the animals to perform the reflex test.

Cliff avoidance test. Cliff avoidance tests were conducted by placing rat pups on the edge of a platform (30 cm × 30 cm × 30 cm), with their forepaws and chest extending over the edge. The latency of the rats to turn away or retreat from the edge was recorded. We scored the test: 0 for no movement or falling off the edge, 1 for attempts to move away from the cliff but with hanging limbs, and 2 for successful movement away from the cliff. A maximum time of 30 s was given for the animals to perform the test. 

Huddling behavior test (HBT). Rat pups were carefully removed from their dam and uniformly spaced in the arena in a radial position at room temperature. They were free to move and huddle for 10 min. At the end, we hand-scored the aggregon pattern. Each possible combination of numbers of bodies in contact was an aggregon. We had 15 aggregons that could be formed by seven individuals, and each aggregon pattern corresponded to an index. The chamber was cleaned and wiped with H_2_O and 70% ethanol between each session.

### 4.10. Statistical Analysis

No statistical methods predetermined sample sizes. We ensured consistency by repeating trials in different cell cultures from at least three animals per condition. Using GraphPad Prism 8, we assessed normal distribution with the Shapiro–Wilk test. For normally distributed data, we used one-way ANOVA with the Holm–Sidak post-hoc test; for non-normal data, we used the Kruskal–Wallis test with Dunn’s post-hoc test. The results are represented as the mean ± standard deviation (SD). In figure legends, N is the number of cultures or dams and n is the number of cells or pups per condition. Statistical analyses were conducted on the data from all cells/pups (n).

## 5. Conclusions

In conclusion, our findings underscore the critical role of timing and developmental stages in elucidating how both forms of BDNF influence the regulation of KCC2 cell membrane trafficking. Particularly, our data strongly suggest that proBDNF delays the GABA shift polarity, thereby maintaining neurons in an immature state. This consequence could be linked to the behavioral deficits observed in electroporated rats. These actions carry significant implications for cognitive processes and neural circuitry, providing insights into the intricate interplay between neurotrophic factors and neuronal functions. Furthermore, a deeper investigation into the specific mechanisms through which BDNF modulates GABA function and chloride homeostasis during development is imperative. This understanding holds potential implications for our comprehension of neurodevelopmental processes and could pave the way for the development of targeted therapies benefiting individuals with brain disorders.

## Figures and Tables

**Figure 1 ijms-25-06253-f001:**
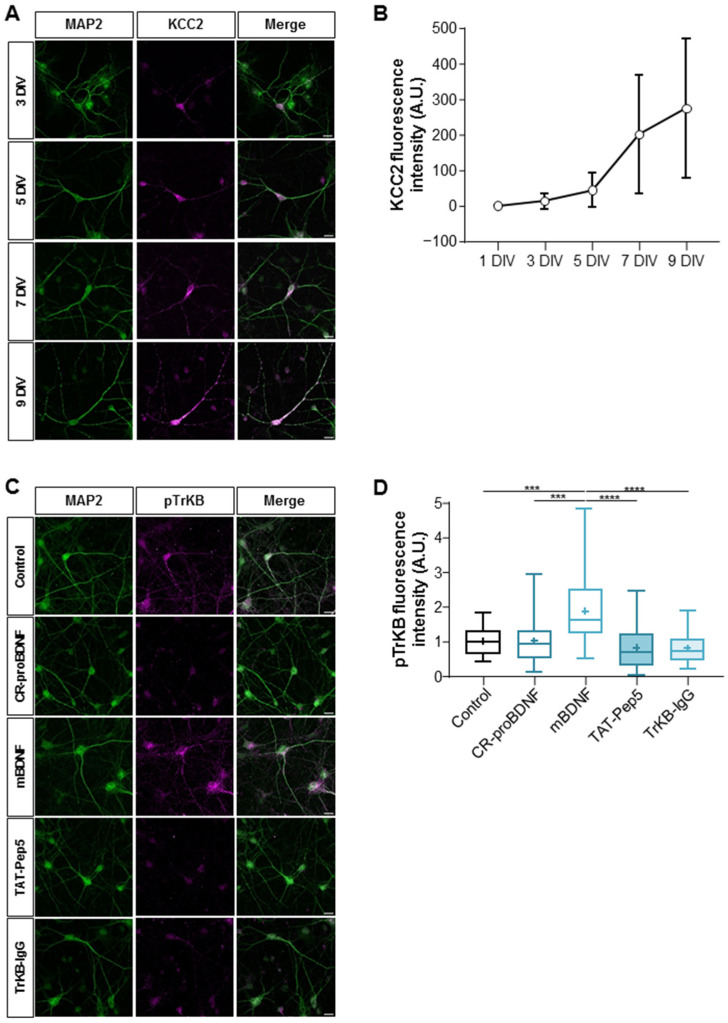
Developmental profile of KCC2 expression and BDNF signaling pathway activation in maturing cultured hippocampal neurons. (**A**) Representative images of immunofluorescence showing MAP2 (green) and KCC2 (purple) expression in hippocampal neurons from 3 DIV to 9 DIV. Scale bar: 10 μm. (**B**) The graph shows that the fluorescence intensity of KCC2 expression increased over time from 1 to 9 DIV. N represents the number of cultures = 3; n represents the number of cells = 35 for 1 DIV; n = 45 for 3 DIV; n = 32 for 5 DIV; n = 45 for 7 DIV; n = 35 for 9 DIV. For each condition, the mean ± SD was represented. (**C**) Representative images of immunofluorescence showing MAP2 (green) and pTrkB (purple) expression in neurons from 6–7 DIV in control and Cr-proBDNF-, mBDNF-, TAT-Pep5-, and TrKB-IgG-treated neurons. Scale bar: 10 μm. (**D**) Box plot showing the quantification of pTrKB fluorescence intensity in the indicated conditions. N = 3; n = 36 for control; n = 38 for CR-proBDNF; n = 41 for mBDNF; n = 37 for TAT-Pep5; n = 45 for TrKB-IgG. Box plots show 25th, 50th, and 75th percentiles, minimum and maximum as whiskers, with the mean indicated as “+”. Compared using the Kruskal–Wallis test followed by Dunn’s post hoc test, *** *p* = 0.0008 for mBDNF vs. control, *** *p* = 0.0001 for mBDNF vs. CR-proBDNF, **** *p* < 0.0001 for mBDNF vs. TrKB-IgG, **** *p* < 0.0001 for mBDNF vs. TAT-Pep5.

**Figure 2 ijms-25-06253-f002:**
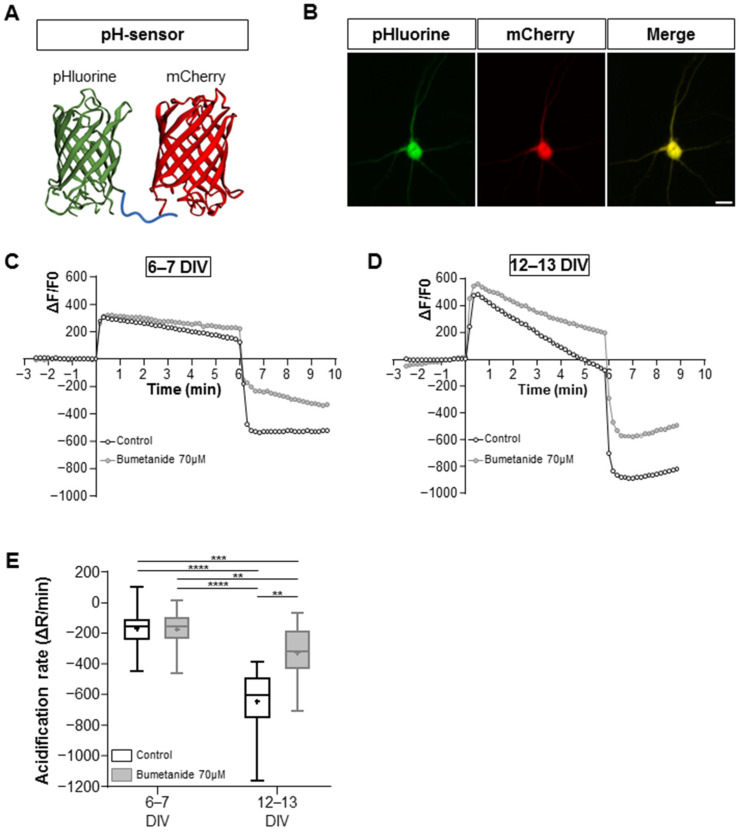
Functional activity of KCC2 in maturing cultured hippocampal neurons. (**A**) A 3D structure of the pH sensor, composed of pH-sensitive fluorine in green and pH-insensitive mCherry in red. (**B**) Representative images of immunofluorescence illustrating the pH sensor (green) in a transfected neuron (red). Scale bar: 150 µm. (**C**,**D**) Graph showing the ΔF/F of recorded neurons in control and bumetanide 70 µM conditions at 6–7 DIV (**C**) and 12–13 DIV (**D**). Neurons were perfused with a 10 mM NH_4_Cl-containing solution at t = 0. Swiftly after NH_4_Cl application, the intracellular space alkalized as NH3 diffused through the plasma membrane. KCC2 began to transport NH4^+^ into the cell, acidifying the intracellular space. The rate of pHi changes during NH4^+^ transportation, visualized by the fluorescence intensity of pHluorin, was dependent on the ion transportation activity of KCC2. (**E**) The acidification rate of neurons in the indicated conditions is visualized as box plots; N = 9 and n = 63 for control 6–7 DIV; N = 5 and n = 31 for bumetanide 70 µM 6–7 DIV; N = 5 and n = 26 for control 12–13 DIV; N = 5 and n = 30 for bumetanide 70 µM 12–13 DIV. Box plots show 25th, 50th, and 75th percentiles, minimum and maximum as whiskers, with the mean indicated as “+”. Compared using Kruskal–Wallis test followed by Dunn’s post hoc test, **** *p* < 0.0001 for control 6–7 DIV vs. control 12–13 DIV, *** *p* = 0.001 for control 6–7 DIV vs. bumetanide 70 µM 12–13 DIV, **** *p* < 0.0001 bumetanide 70 µM 6–7 DIV vs. control 12–13 DIV, *** p* = 0.0061 bumetanide 70 µM 6–7 DIV vs. bumetanide 70 µM 12–13 DIV, ** *p* = 0.0012 for control 12–13 DIV vs. bumetanide 70 µM 12–13 DIV.

**Figure 3 ijms-25-06253-f003:**
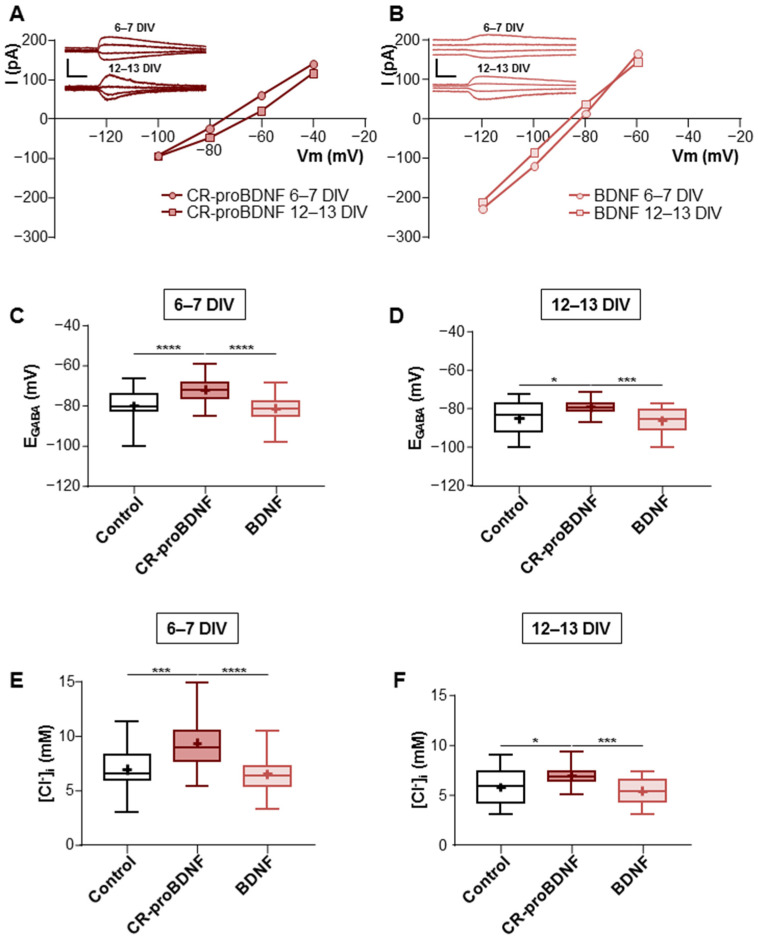
CR-proBDNF promotes GABA depolarization polarity in hippocampal neurons. (**A**) Gramicidin-perforated patch-clamp recording current–voltage (I–V) relationships for isoguvacine-response currents in hippocampal primary cultures at 6–7 and 12–13 DIV transfected with CR-proBDNF-mCherry. Inserts depict the isoguvacine currents at both ages. Scale bars: 200 ms, 200 pA. (**B**) I-V and inserts depict the BDNF-mCherry transfected neurons. Scale bars: 200 ms, 200 pA. (**C**,**D**) Box plots of E_GABA_ for the indicated conditions at 6–7 and 12–13 DIV. (**E**,**F**) Box plots of [Cl^−^]_i_ for the indicated conditions at 6–7 and 12–13 DIV. For 6–7 DIV: N = 10 and n = 29 for control; N = 8 and n = 40 for CR-proBDNF; N = 10 and n = 34 for BDNF. For 12–13 DIV: N = 7 and n = 22 for control; N = 5 and n = 26 for CR-proBDNF; N = 8 and n = 25 for BDNF. Box plots show 25th, 50th, and 75th percentiles, minimum and maximum as whiskers, with the mean indicated as “+”. Compared using the Kruskal–Wallis test followed by Dunn’s post hoc test. At 6–7 DIV: *** *p* = 0.0002 for E_GABA_ and [Cl^−^]_i_ of CR-proBDNF vs. control, **** *p* < 0.0001 for E_GABA_ and [Cl^−^]_i_ of CR-proBDNF vs. BDNF. At 12–13 DIV: ** p* = 0.0295 for E_GABA_ of CR-proBDNF vs. control, *** *p* = 0.0008 for E_GABA_ of CR-proBDNF vs. BDNF. Compared using the one-way ANOVA test followed by Holm–Sidak’s post hoc test at 12–13 DIV for [Cl^−^]_i_, * *p* = 0.0104 for CR-proBDNF vs. control, *** *p* = 0.0007 for CR-proBDNF vs. BDNF.

**Figure 4 ijms-25-06253-f004:**
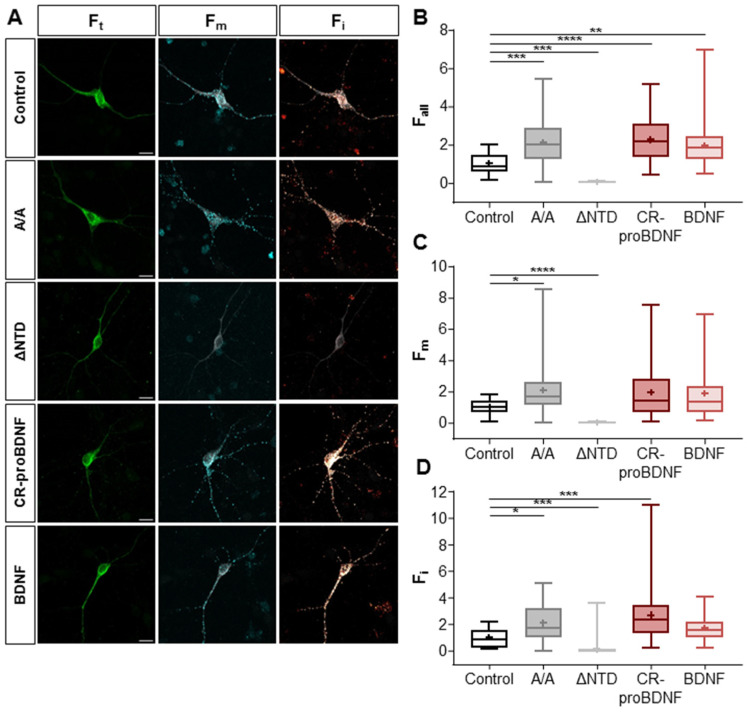
CR-proBDNF increases the internalization of KCC2. (**A**) Representative images of immunofluorescence illustrating total (F_t_; green) membrane (F_m_; blue) and internalized (F_i_; red) pools of KCC2 with an external tag (KCC2-pH_ext_) for the control, CR-ProBDNF-mCherry, BDNF-mCherry, A/A-KCC2-pH_ext_, and ΔNTD-KCC2-pH_ext_ conditions in hippocampal primary culture neurons. Scale bar: 10 μm. Box plots of (**B**) all clusters (F_all_) and (**C**) membranes (F_m_) and (**D**) internalized pool (F_i_) fluorescence normalized per cell in cultured neurons of the indicated conditions. N = 4; n = 27 for control; n = 56 for A/A; n = 39 for ΔNTD; n = 57 for CR-proBDNF; n = 55 for BDNF. Box plots show 25th, 50th, and 75th percentiles, minimum and maximum as whiskers, with the mean indicated as “+”. Compared using the Kruskal–Wallis test followed by Dunn’s post hoc test. F_all_: *** *p* = 0.0005 for control vs. A/A, *** *p* = 0.0009 for control vs. ΔNTD, **** *p* < 0.0001 for CR-proBDNF vs. control, ** *p =* 0.0026 for BDNF vs. control; F_m_: * *p* = 0.019 for control vs. A/A, **** *p* < 0.0001 for control vs. ΔNTD; F_i_: * *p* = 0.015 for control vs. A/A, *** *p* = 0.0003 for control vs. ΔNTD, *** *p* = 0.0001 for CR-proBDNF vs. control, *p* = 0.074 for BDNF vs. control.

**Figure 5 ijms-25-06253-f005:**
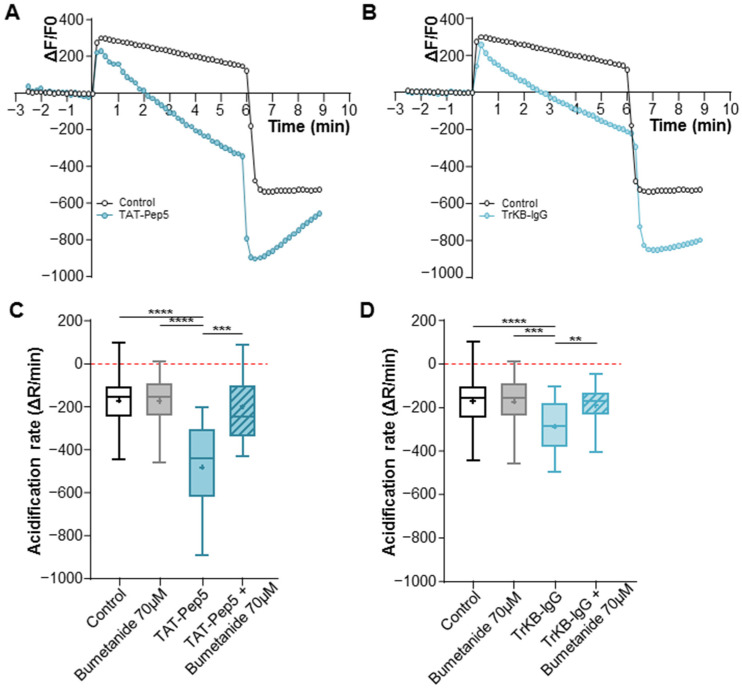
The activity of KCC2 is increased with the inhibitors of both forms of BDNF. (**A**) Graph showing the ΔF/F of recorded neurons at 6–7 DIV for the control and TAT-Pep5 conditions. (**B**) Graph showing the ΔF/F of recorded neurons at 6–7 DIV for the control and TrKB-IgG conditions. (**C**,**D**) The acidification rate of neurons in the indicated conditions at 6–7 DIV is visualized as box plots. N = 9 and n = 63 for control; N = 5 and n = 31 for bumetanide 70 µM; N = 6 and n = 20 for TAT-Pep5; N = 4 and n = 26 for TAT-Pep5 + bumetanide 70 µM; N = 6 and n = 24 for TrKB-IgG; N = 4 and n = 26 for TrKB-IgG + bumetanide 70 µM. Box plots show 25th, 50th, and 75th percentiles, minimum and maximum as whiskers, with the mean indicated as “+”. Compared using the Kruskal–Wallis test followed by Dunn’s post hoc test, **** *p* < 0.0001 for TAT-Pep5 vs. control, **** *p* < 0.0001 for TAT-Pep5 vs. bumetanide 70 µM, *** *p* = 0.0008 for TAT-Pep5 vs. TAT-Pep5 + bumetanide 70 µM. Compared using the one-way ANOVA test followed by Holm–Sidak’s post hoc test, **** *p* < 0.0001 for TrKB-IgG vs. control, *** *p* = 0.0006 for TrKB-IgG vs. bumetanide 70 µM, ** *p* = 0.0065 for TrKB-IgG vs. TrKB-IgG + bumetanide 70 µM.

**Figure 6 ijms-25-06253-f006:**
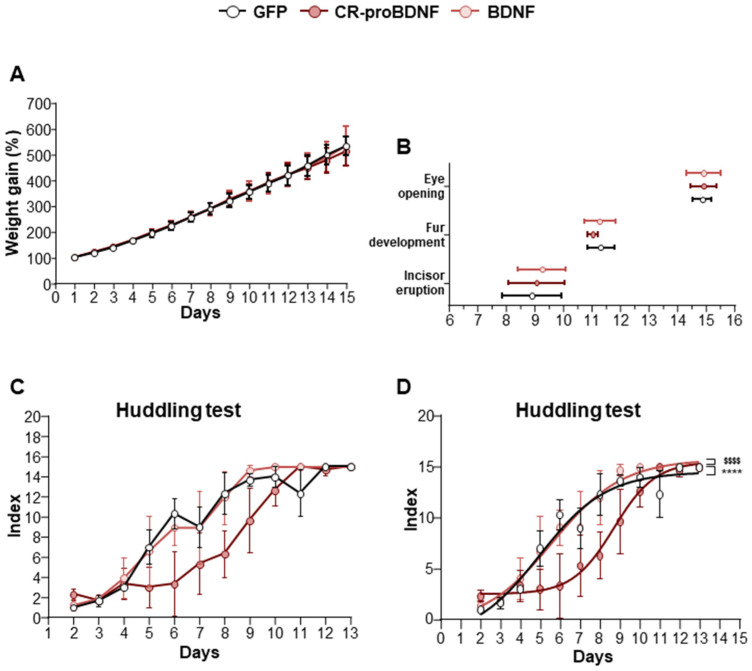
CR-proBDNF-electroporated rats exhibited normal physical landmarks and behavioral deficits. (**A**) Percentage of weight gain from PN1 to PN15 in GFP and CR-proBDNF- and BDNF-electroporated rats. (**B**) Physical landmarks: eye pening, fur development, and incisor eruption in the indicated conditions. n represents the number of pups for each condition = 32 for GFP; n = 30 for CR-proBDNF; n = 28 for BDNF. (**C**) Index of aggregon patterns from PN2 to PN13 in the indicated conditions. (**D**) Nonlinear regression fit model of data from (**C**). N = 3 dams of rats for each condition. For each group, the mean ± SD was represented. **** *p* < 0.0001 and F = 16.6 for CR-proBDNF vs. GFP; $$$$ *p* < 0.0001 and F = 14.7 for CR-proBDNF vs. BDNF.

**Figure 7 ijms-25-06253-f007:**
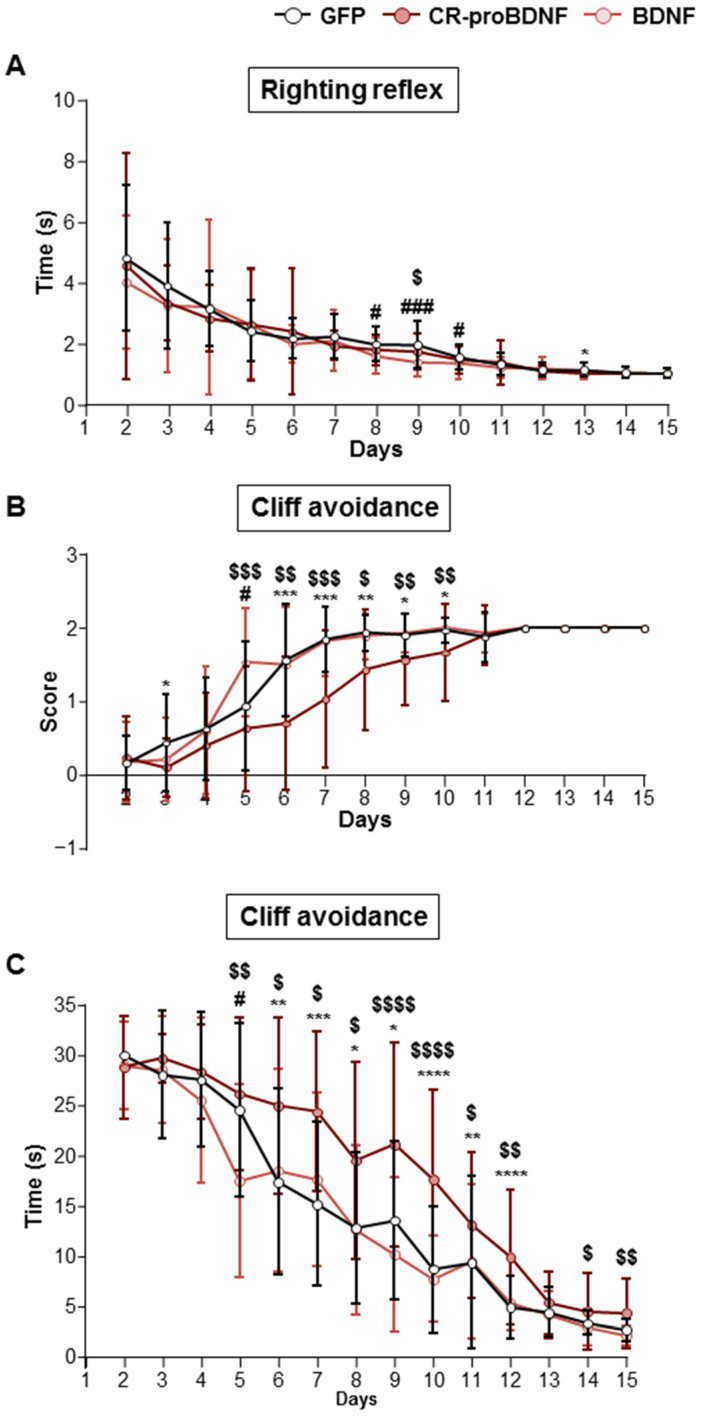
CR-proBDNF-electroporated rats exhibited deficits in sensory maturation. (**A**) Time spent to complete the righting reflex from PN2 to PN15 for GFP and CR-proBDNF- and BDNF-electroporated rats. Compared using the Kruskal–Wallis test followed by Dunn’s post hoc test. PN8: # *p* = 0.02 for BDNF vs. GFP; PN9: ### *p* = 0.0006 for BDNF vs. GFP, $ *p* = 0.03 for CR-proBDNF vs. BDNF; PN10: # *p* = 0.017 for BDNF vs. GFP; PN13: * *p* = 0.016 for CR-proBDNF vs. GFP. (**B**) Score of cliff avoidance success from PN2 to PN15 in the indicated conditions. (**C**) Time spent to avoid the cliff from PN2 to PN15 in the indicated conditions. n = 32 for GFP; n = 30 for CR-proBDNF; n = 28 for BDNF. For each group, the mean ± SD is represented. Compared using the Kruskal–Wallis test followed by Dunn’s post hoc test. * *p* < 0.05, ** *p* < 0.01, *** *p* < 0.001, and **** *p* < 0.0001 for CR-proBDNF vs. GFP. # *p* < 0.05 for BDNF vs. GFP. $ *p* < 0.05, $$ *p* < 0.01, $$$ *p* < 0.001, and $$$$ *p* < 0.0001 for CR-proBDNF vs. BDNF.

**Table 1 ijms-25-06253-t001:** Score of cliff avoidance test.

Days/Condition	GFP	CR-proBDNF	BDNF
2	0.16 ± 0.37	0.23 ± 0.57	0.18 ± 0.55
3	0.44 ± 0.67	0.1 ± 0.4 *	0.21 ± 0.57
4	0.63 ± 0.7	0.4 ± 0.72	0.61 ± 0.88
5	0.94 ± 0.88	0.63 ± 0.85	1.54 ± 0.74 #
6	1.56 ± 0.76	0.7 ± 0.92 ***	1.5 ± 0.79 $$
7	1.84 ± 0.45	1.03 ± 0.93 ***	1.82 ± 0.48 $$$
8	1.94 ± 0.25	1.43 ± 0.82 **	1.9 ± 0.31 $
9	1.91 ± 0.3	1.6 ± 0.62 *	1.93 ± 0.26 $$
10	1.97 ± 0.18	1.7 ± 0.66 *	2 ± 0 $$
11	1.88 ± 0.34	1.9 ± 0.4	1.93 ± 0.26
12	2 ± 0	2 ± 0	2 ± 0
13	2 ± 0	2 ± 0	2 ± 0
14	2 ± 0	2 ± 0	2 ± 0
15	2 ± 0	2 ± 0	2 ± 0

Data are presented as mean ± SD, compared using the Kruskal–Wallis test followed by Dunn’s post hoc test. * *p* < 0.05, ** *p* < 0.01, and *** *p* < 0.001 for CR-proBDNF vs. GFP. # *p* < 0.05 for BDNF vs. GFP. $ *p* < 0.05, $$ *p* < 0.01, and $$$ *p* < 0.001 for CR-proBDNF vs. BDNF.

**Table 2 ijms-25-06253-t002:** Time spent to accomplish the cliff avoidance test.

Days/Condition	GFP	CR-proBDNF	BDNF
2	30 ± 0	28.7 ± 5.08	28.96 ± 4.31
3	28 ± 6.37	29.6 ± 2.37	28.54 ± 5.39
4	27.53 ± 6.7	28.23 ± 4.75	25.54 ± 8.24
5	24.5 ± 8.67	26.03 ± 7.62	17.5 ± 9.68 $$ #
6	17.4 ± 9.23	24.9 ± 8.79 **	18.54 ± 10.11
7	15.16 ± 8.15	24.3 ± 7.95 ***	17.64 ± 8.63 $
8	12.78 ± 7.51	19.4 ± 9.77 *	12.61 ± 8.45 $
9	13.5 ± 7.93	21.03 ± 10.16 *	10.18 ± 7.65 $$$$
10	8.7 ± 6.3	17.5 ± 8.95 ****	7.71 ± 4.31 $$$$
11	9.31 ± 8.59	13 ± 7.27 **	9.5 ± 7.71 $
12	4.9 ± 3.07	9.83 ± 6.74 ****	5.32 ± 2.68 $$
13	4.4 ± 2.47	5.3 ± 3.01	4.14 ± 2.37
14	3.31 ± 1.18	4.4 ± 3.78	2.9 ± 1.75 $
15	2.63 ± 1.16	4.23 ± 3.45	2.11 ± 0.95 $$

Data are presented as mean ± SD, compared using the Kruskal–Wallis test followed by Dunn’s post hoc test. * *p* < 0.05, ** *p* < 0.01, *** *p* < 0.001, and **** *p* < 0.0001 for CR-proBDNF vs. GFP. # *p* < 0.05 for BDNF vs. GFP. $ *p* < 0.05, $$ *p* < 0.01, and $$$$ *p* < 0.0001 for CR-proBDNF vs. BDNF.

## Data Availability

Data supporting the findings of this study are available in this manuscript. All other data that support the findings of this study are available from the corresponding authors upon reasonable request.

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
