# Peer review of "Regulation of Neuronal Chloride Homeostasis by Pro- and Mature Brain-Derived Neurotrophic Factor (BDNF) via KCC2 Cation–Chloride Cotransporters in Rat Cortical Neurons"

_ijms, 2024, doi:10.3390/ijms25116253_

Round 1

Reviewer 1 Report

Comments and Suggestions for Authors

In the manuscript submitted for review, the Authors analyzed neuronal chloride homeostasis by pro- and mature brain-derived neurotrophic factor (BDNF) via KCC2 cation-chloride cotransporters in rat cortical neurons. I find the subject of the manuscript interesting and "current", and the entire work is well thought out. The Authors put a lot of work into preparing this interesting work. The very carefully prepared engravings undoubtedly attract the reader's attention.

my comments:

1. Unfortunately, the rather long introduction does not contain a clearly and precisely formulated hypothesis;

2. perfect photographs contain a laconic description of what they show - the legend could be supplemented with information about what they contain (e.g. FIGURE 1 A and C);

Author Response

 We thank the reviewer for his(her) positive comments regarding our study and for his(her) constructive criticism.

  1. Unfortunately, the rather long introduction does not contain a clearly and precisely formulated hypothesis;

We agree with the referee's comment and, to clarify the purpose of this study, we have added the following sentence to the introduction (L93-L97):

“In this study, we aim to investigate the respective roles of both forms of BDNF in regulating chloride homeostasis and GABAergic inhibitory strength during early developmental stages. We will achieve this by examining the functional expression of KCC2 in cultured rat hippocampal neurons. Additionally, we will explore the behavioral consequences of proBDNF expression using electroporated rat pups.”

  1. perfect photographs contain a laconic description of what they show - the legend could be supplemented with information about what they contain (e.g. FIGURE 1 A and C);

We agree with the referee's comment. In response, we have added information to better describe the immunofluorescence panels:

Figure 1; L128: “(A) Representative images of immunofluorescence showing MAP2 (green) and KCC2 (purple) expression in hippocampal neurons from 3 DIV to 9 DIV.”

L133: “(C) Representative images of immunofluorescence showing MAP2 (green) and pTrkB (purple) expression in neurons from 6-7 DIV…”

Figure 2; L164: “(B) Representative images of immunofluorescence illustrating the pH-sensor (green) in a transfected neuron (red).”

Figure 4; L263: “(A) Representative pseudo-colored images of immunofluorescence illustrating total (Ft;; green) membrane (Fm; blue) and internalized (Fi; red) pools of KCC2 with an external tag (KCC2-pHext)...”

Reviewer 2 Report

Comments and Suggestions for Authors

This is very well written manuscript. This study conducted a series of in vitro and in vivo experiments to address a significant scientific question: the role of proBDNF and mBDNF in regulation of neuronal chloride homeostasis. The experiments were very well designed. The study provided solid evidence to support the role of BDNF signaling in regulating chloride transport through the modulation of KCC2. There are several minor comments:

1, Article title, line 4: since all in vitro experiments were conducted in rat hippocampal neurons to examine the interplays between BDNF, KCC2, chloride homeostasis, and inhibitory synaptic transmission, it is suitable to emphasize on “rat cortical neurons” in the title.

2. Figure legends: although meaning of “N’ and ‘n” have been explained in the Methods, it is better to explain them at least in the legend of Figure 1.

3. line 215: Was this experiment also conducted in the hippocampal neurons? Make it clear.

4. Statistical analysis: I assumed that analysis was conducted on data from all cells/pups (n), but not on average data of cultures/dams (N). Make this clear.

Author Response

 We thank the reviewer for his(her) positive comments regarding our study and for his(her) constructive criticism.

1, Article title, line 4: since all in vitro experiments were conducted in rat hippocampal neurons to examine the interplays between BDNF, KCC2, chloride homeostasis, and inhibitory synaptic transmission, it is suitable to emphasize on “rat cortical neurons” in the title.

We understand the referee's remark. In the title, we used the term "cortical" in a general sense to refer to cortical structures, as the hippocampus is an extension of the temporal part of the cerebral cortex. In the abstract, we clearly distinguish between cultures of hippocampal neurons and experiments on electroporated cortices of rat pups.

  1. Figure legends: although meaning of “N’ and ‘n” have been explained in the Methods, it is better to explain them at least in the legend of Figure 1.

We agree with the referee's comment. In response, we have added the following information:

Figure 1(L134): “N represents the number of cultures =3; n represents the number of cells =35 for 1 DIV.”

Figure 6 (L336): “n represents the number of pups for each condition =32 for GFP; n=30 for CR-proBDNF; n=28 for BDNF pups.”

  1. line 215: Was this experiment also conducted in the hippocampal neurons? Make it clear.

We thank the reviewer for this comment. We have modified the sentence accordingly (L231-L234): “We, therefore, assessed whether CR-proBDNF and BDNF could regulate KCC2 stability at the cell surface of hippocampal neurons by measuring KCC2 expression in different cell compartments at 9 DIV.”

  1. Statistical analysis: I assumed that analysis was conducted on data from all cells/pups (n), but not on average data of cultures/dams (N). Make this clear.

We agree with the referee's comment. In response, we have added the following information:

L622: “Statistical analysis was done on the data from all cells/pups (n).”